# MakeAnything: Harnessing Diffusion Transformers for Multi-Domain Procedural Sequence Generation

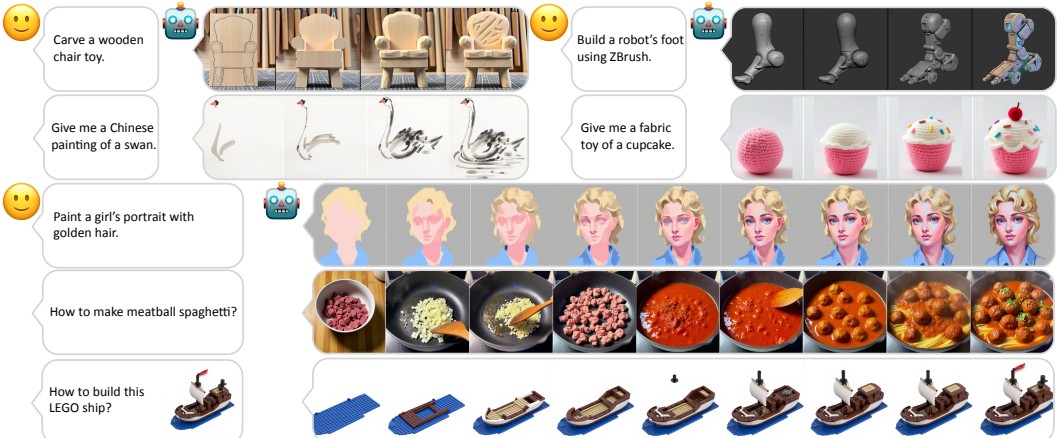

Figure 1: We introduce MakeAnything, a tool that realistically and logically generates step-by-step procedural tutorial for activities such as painting, crafting, and cooking, based on text descriptions or conditioned images.

## ABSTRACT

A hallmark of human intelligence is the ability to create complex artifacts through structured multi-step processes. Generating procedural tutorials with AI is a long-standing but challenging goal, facing three key obstacles: (1) scarcity of multi-task procedural datasets, (2) maintaining logical continuity and visual consistency between steps, and (3) generalizing across multiple domains. To address these challenges, we propose a multi-domain dataset covering 21 tasks with over 24,000 procedural sequences. Building upon this foundation, we introduce MakeAnything, a framework based on the diffusion transformer (DIT), which leverages fine-tuning to activate the in-context capabilities of DIT for generating consistent procedural sequences. We introduce asymmetric low-rank adaptation (LoRA) for image generation, which balances generalization capabilities and task-specific performance by freezing encoder parameters while adaptively tuning decoder layers. Additionally, our ReCraft model enables image-to-process generation through spatiotemporal consistency constraints, allowing static images to be decomposed into plausible creation sequences. Extensive experiments demonstrate that MakeAnything surpasses existing methods, setting new performance benchmarks for procedural generation tasks.

## 1 INTRODUCTION

A defining characteristic of human intelligence—and a key differentiator from other species—is the capacity to create complex artifacts through structured step-by-step processes. In computer vision, generating such procedural sequences for tasks like painting, crafting, product design, and culinary arts remains a significant challenge. The core difficulty lies in producing multi-step sequences that

maintain logical continuity and visual consistency, requiring models to both capture intricate visual features and understand causal relationships between steps. This challenge becomes particularly pronounced when handling diverse domains and styles without compromising generation quality—a problem space that remains underexplored.

Existing research primarily focuses on decomposing painting processes, with early methods employing reinforcement learning/optimization algorithms through stroke-based rendering to approximate target images. Subsequent works like ProcessPainter Song et al. (2024a) and PaintsUndo Team (2024) utilize temporal models on synthetic datasets, while Inverse Painting Chen et al. (2024)redicts the order of human painting, generating the painting process by region. However, these approaches remain limited to single-task scenarios and exhibit poor cross-domain generalization. Furthermore, ProcessPainter's Animatediff-based framework constrains modifications to minor motion adjustments, making it unsuitable for categories requiring structural transformations (e.g., recipes or crafts). Although Diffusion Transformer (DIT) Peebles & Xie (2023)-based video generation models can produce long sequences, their effectiveness is hindered by distribution shifts in training data when generating complex procedural workflows.

We posit that replicating human creative intelligence requires both high-quality multi-task procedural data and advanced methodology design.. To this end, we curate a comprehensive multi-domain dataset spanning 21 categories (including painting, crafts, SVG design, LEGO assembly, and cooking) with over 24,000 procedurally annotated sequences—the largest such collection for step-by-step creation tasks. Methodologically, we propose MakeAnything, a novel framework that harnesses the in-context capabilities of Diffusion Transformers (DIT) through LoRA fine-tuning to generate high-quality instructional sequences.

Addressing the challenge of severe data scarcity (some categories have as few as 50 data entries.) and imbalanced distributions, we employ an asymmetric low-rank adaptation (LoRA)Zhu et al. (2024); Hu et al. (2022) strategy for image generation. This approach combines a pretrained encoder on large-scale data with a task-specific fine-tuned decoder, achieving an optimal balance between generalization and domain-specific performance.

To address practical needs for reverse-engineering creation processes, we develop the ReCraft Model—an efficient controllable generation method that decomposes static images into step-by-step procedural sequences. Building upon the pretrained Flux model with minimal architectural modifications, ReCraft introduces an image-conditioning mechanism where clean latent tokens from the target image (encoded via VAE) guide the denoising of noisy intermediate frames through multi-modal attention. Remarkably, this lightweight adaptation enables efficient training with limited data—ReCraft achieves robust performance with just hundreds or even dozens of process sequences per task. During inference, the model recursively predicts preceding frames over concatenated latent representations, effectively reconstructing the creation history from static artworks.

In summary, our contributions are as follows:

1. **Unified Procedural Generation Framework**: We introduce MakeAnything, the first DIT-based architecture enabling cross-domain procedural sequence synthesis, supporting both text-to-process and image-to-process generation paradigms.

2. **Technical Innovations**: We employ an asymmetric LoRA architecture for cross-domain generalization and the ReCraft Model for image-conditioned process reconstruction with limited training data.

3. **Dataset Contribution**: We propose a multi-domain procedural dataset (21 categories, 24K+ sequences) with hierarchical annotations, significantly advancing research in procedural understanding and generation.

## 2 RELATED WORK

### 2.1 DIFFUSION MODELS

Diffusion probability models (Song et al., 2020; Ho et al., 2020) are advanced generative models that restore original data from pure Gaussian noise by learning the distribution of noisy data at various levels of noise. Their powerful capability to adapt to complex data distributions has led diffusion models to achieve remarkable success across several domains including image synthesis (Rombach

et al., 2022; Peebles & Xie, 2023), image editing (Brooks et al., 2023; Hertz et al., 2022; Zhang et al., 2024c;d; Yang et al., 2024; Li et al., 2024), and video gneration (Guo et al., 2023; Blattmann et al., 2023; Song et al., 2024a), evaluation (Song et al., 2024b). Stable Diffusion (Rombach et al., 2022) (SD), a notable example, utilizes a U-Net architecture and extensively trains on large-scale text-image datasets to iteratively generate images with impressive text-to-image capabilities. The Diffusion Transformer (DiT) model Peebles & Xie (2023), employed in architectures like FLUX.1 AI (2024), Stable Diffusion 3 Esser et al. (2024), and PixArt (pixart), uses a transformer as the denoising network to iteratively refine noisy image tokens. Customized generation methods enable flexible customization of concepts and styles by fine-tuning U-Net (Ruiz et al., 2023) or certain parameters (Hu et al., 2022; Kumari et al., 2023), alongside trainable tokens. Training-free customization methods (Ye et al., 2023; Zhang et al., 2024a;b; Zeng et al., 2023) leverage pre-trained CLIP (Radford et al., 2021) encoders to extract image features for efficient customized generation.

## 2.2 CONTROLLABLE GENERATION IN DIFFUSION MODELS

Controllable generation has been extensively studied in the context of diffusion models. Text-to-image models Ho et al. (2020); Song et al. (2020) have established a foundation for conditional generation, while various approaches have been developed to incorporate additional control signals such as images. Notable methods include ControlNet Zhang & Agrawala (2023), enabling spatially aligned control in diffusion models, and T2I-Adapter Mou et al. (2023), which improves efficiency with lightweight adapters. UniControl Zhao et al. (2023) uses Mixture-of-Experts (MoE) to unify different spatial conditions, further reducing model size. However, these methods rely on spatially adding condition features to the denoising network's hidden states, inherently limiting their effectiveness for spatially non-aligned tasks like subject-driven generation. IP-Adapter Ye et al. (2023) addresses this by introducing cross-attention through an additional encoder. Based on the DiT architecture, OminiControl Tan et al. (2024) proposes a unified solution that is applicable to both spatially aligned and non-aligned tasks by concatenating condition tokens with noise tokens.

## 2.3 PROCEDURAL SEQUENCES GENERATION

Generating the creation process of paintings or handicrafts is something that has always been desired but is difficult to achieve. The problem of teaching machines "how to paint" has been thoroughly explored within stroke-based rendering (SBR), focusing on recreating non-photorealistic imagery through strategic placement and selection of elements like paint strokes Hertzmann (2003). Early SBR methods included greedy searches or required user input Haeberli (1990); Litwinowicz (1997), while recent advancements have utilized RNNs and RL to sequentially generate strokes Ha & Eck (2017); Zhou et al. (2018); Xie et al. (2013). Adversarial training has also been introduced as an effective way to produce non-deterministic sequences Nakano (2019). Techniques like Stylized Neural Painting Kotovenko et al. (2021) have advanced stroke optimization, which can be integrated with neural style transfer. The field of vector graphic generation employs similar techniques Frans et al. (2022); Song et al. (2023); Song (2022); Song & Zhang (2022). However, these methods differ greatly from human creative processes due to variations in artists' styles and subjects. Inverse Painting Chen et al. (2024) achieves realistic painting process simulation by predicting the painting order and implementing image segmentation. ProcessPainter Song et al. (2024a) and Paints Undo Team (2024) method fine-tunes diffusion models using data from artists' painting processes to learn their true distributions, enabling the generation of painting processes in multiple styles.

## 3 METHOD

This section first reviews diffusion transformers (Sec. 3.1), then outlines the MakeAnything architecture (Sec. 3.2). We introduce asymmetric LoRA for procedural learning (Sec. 3.3), followed by the ReCraft model for image-conditioned sequence generation (Sec. 3.4), and conclude with our proposed dataset (Sec. 3.5).

### 3.1 PRELIMINARY

The Diffusion Transformer (DiT) model, uses a transformer as the denoising network to iteratively refine noisy image tokens. A DiT model processes two types of tokens: noisy image tokens $z \in \mathbb{R}^{N \times d}$ and text condition tokens $c_T \in \mathbb{R}^{M \times d}$, where $d$ is the embedding dimension, and $N$ and $M$ are the

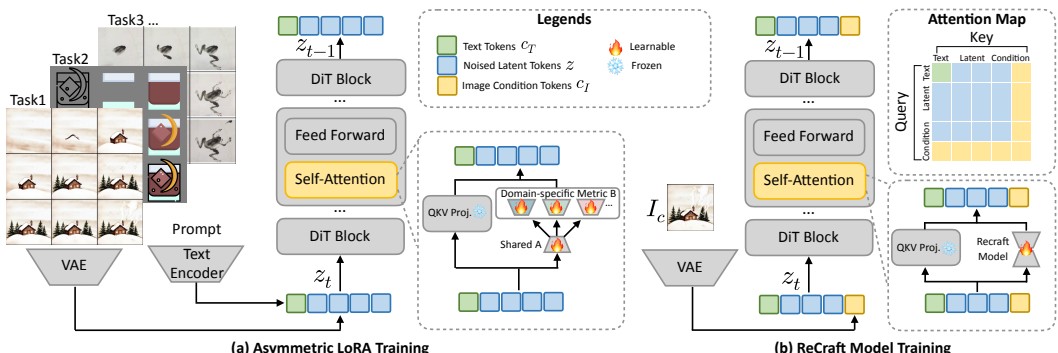

Figure 2: The MakeAnything framework comprises two core components: (1) an Asymmetric LoRA module that generates diverse creation processes from text prompts through asymmetric LoRA, and (2) the ReCraft Model, which constructs an image-conditioned base model by merging pretrained LoRA weights with the Flux foundation model, enabling process prediction via injected visual tokens.

number of image and text tokens. Throughout the network, these tokens maintain consistent shapes as they pass through multiple transformer blocks.

In FLUX.1, each DiT block consists of layer normalization followed by Multi-Modal Attention (MMA) Pan et al. (2020), which incorporates Rotary Position Embedding (RoPE) Su et al. (2024) to encode spatial information. For image tokens $z$, RoPE applies rotation matrices based on the token's position $(i, j)$ in the 2D grid:

$$z_{i,j} \rightarrow z = z_{i,j} \cdot R(i, j), \tag{1}$$

where $R(i, j)$ is the rotation matrix at position $(i, j)$. Text tokens $c_T$ undergo the same transformation with their positions set to $(0, 0)$.

The multi-modal attention mechanism then projects the position-encoded tokens into query $Q$, key $K$, and value $V$ representations. It enables the computation of attention between all tokens:

$$\text{MMA}([z; c_T]) = \text{softmax}\left(\frac{QK^\top}{\sqrt{d}}\right) V, \tag{2}$$

where $[z; c_T]$ denotes the concatenation of image and text tokens. This formulation enables bidirectional attention.

## 3.2 OVERALL ARCHITECTURE

As shown in Fig. 2, the training of MakeAnything is divided into two stages: First, we train on the MakeAnything dataset using the asymmetric LoRA method, enabling the generation of creative tutorials from text descriptions. Then, the LoRA from this first phase is merged with the Flux base model to form the base model for training the ReCraft Model. The second stage introduces image-conditioned denoising by concatenating condition tokens with noised latents, followed by LoRA-based fine-tuning of the ReCraft model.

## 3.3 ASYMMETRIC LORA FOR PROCEDURAL LEARNING

**Serpentine Sequence Layout.** The core of MakeAnything involves arranging different frames of a sequence into grid and using the in-context capabilities and attention mechanism of DiT to achieve consistent Sequence generation. Tokens within the DiT's attention mechanism tend to focus on spatially adjacent tokens, a tendency that stems from the strong correlations between neighboring image pixels captured during the pre-training of the diffusion model Wan et al. (2024). To enhance the model's learning effectiveness for grid sequences, we propose the Serpentine dataset construction method. As shown in Fig. 3, we arrange sequences of 9 frames and 4 frames in a serpentine pattern to ensure that temporally adjacent frames are also spatially adjacent (either horizontally or vertically adjacent).

**Asymmetric LoRA.** Another challenge is that training a single LoRA on all data leads to difficulties in learning diverse knowledge, while training LoRA on a single type of sequence data results in

overfitting due to the limited quantity of process data for each category. Inspired by HydraLoRA Zhu et al. (2024), we introduce an asymmetric LoRA design for the first time in image generation. This design combines shared knowledge and specialized functionalities by jointly training a shared central matrix $A$ and multiple task-specific matrices $B$, significantly improving multi-task performance.

Each layer of LoRA consists of an $A$ matrix and a $B$ matrix, where $A$ captures general knowledge, and $B$ adapts to specific tasks. The asymmetric LoRA architecture can be formulated as:

$$W = W_0 + \Delta W = W_0 + \sum_{i=1}^{N} \omega_i \cdot B_i A, \tag{3}$$

where $B_i \in \mathbb{R}^{d \times r}$ and the shared matrix $A \in \mathbb{R}^{r \times k}$, $\omega_i$ is the weight of the $i$-th LoRA module. This structure effectively balances generalization and task-specific adaptation, enhancing the model's performance across diverse tasks.

Inference stage, the domain-specific matrix B and the domain-agnostic matrix A are used in combination, balancing generalization capabilities with performance on specific tasks. Our method can also be combined with the stylized LoRA from the Civitai website (which is not trained on procedural sequences), to enhance performance in unseen domains.

**Conditional Flow Matching Loss.** The conditional flow matching loss function is following SD3 Esser et al. (2024), which is defined as follows:

$$L_{CFM} = E_{t,p_t(z|\epsilon),p(\epsilon)} \left[ \|v_\Theta(z, t, c_T) - u_t(z|\epsilon)\|^2 \right] \tag{4}$$

Where $v_\Theta(z, t, c_T)$ represents the velocity field parameterized by the neural network's weights, $t$ is timestep, $c_I$ and $c_T$ are image condition tokens extracted from source image $I_{src}$ and text tokens. $u_t(z|\epsilon)$ is the conditional vector field generated by the model to map the probabilistic path between the noise and true data distributions, and $E$ denotes the expectation, involving integration or summation over time $t$, conditional $z$, and noise $\epsilon$.

### 3.4 ReCraft Model

In practical applications, users not only want to generate creation processes from text but also wish to upload an image and predict the creation process of the existing artwork or handicraft in the picture. For this, we implemented the ReCraft model, which allows users to upload images and generates a sequence of steps highly consistent with the uploaded image.

A major challenge in training the ReCraft model is the limited number of datasets available for each task, which is insufficient to train a controllable plugin like Controlnet or IP-Adapter from scratch. To address this, we innovatively designed the ReCraft model by reusing the pretrained Flux model and making minimal modifications to extend it into an image-conditioned model. Specifically, during training, we input the final frame into a VAE to obtain latent image condition tokens, which are then concatenated with noised latent tokens, using the attention mechanism to provide conditional information for denoising other frames. Notably, the noise addition and removal process are only performed on other frames, while the image condition tokens are clean. During inference, we reconstruct the prior eight steps from the final frame, revealing the object's formation process.

In ReCraft model, multi-modal attention mechanisms are used to provide conditional information for the denoising of other frames.

$$\text{MMA}([z; c_I; c_T]) = \text{softmax}\left(\frac{QK^\top}{\sqrt{d}}\right) V, \tag{5}$$

where $[z; c_I; c_T]$ denotes the concatenation of image and text tokens. This formulation enables bidirectional attention. The conditional flow matching loss with image condition can be defined as:

$$L_{CFM} = E_{t,p_t(z|\epsilon),p(\epsilon)} \left[ \|v_\Theta(z, t, c_I, c_T) - u_t(z|\epsilon)\|^2 \right] \tag{6}$$

Where $v_\Theta(z, t, c_I, c_T)$ represents the velocity field parameterized by the neural network's weights. During inference, Recraft Model predict previous 8 frames based on the final frame. This predicts how the object in the reference image was created step by step.

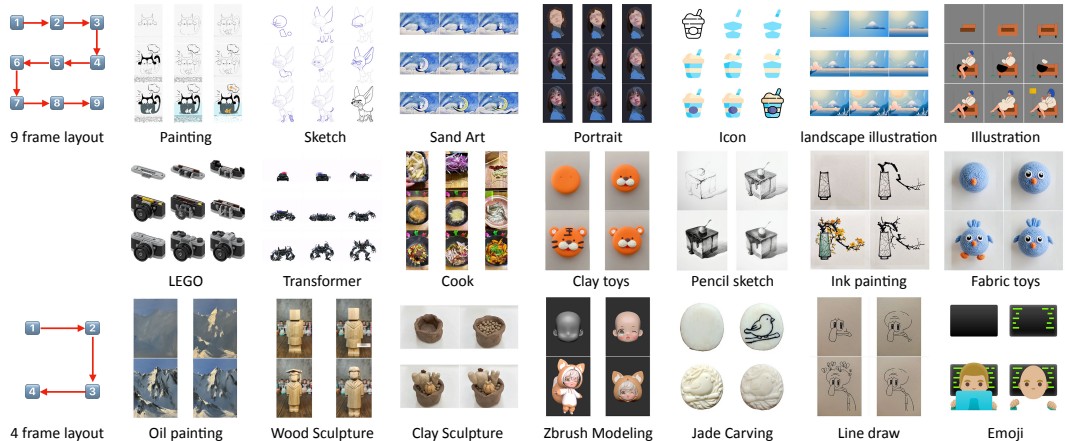

Figure 3: Examples from the MakeAnything Dataset, which consists of 21 tasks with over 24,000 procedural sequences.

### 3.5 MakeAnything Dataset

As shown in Fig. 3, we have collected a multi-task dataset that encompasses tutorials for 21 tasks. We assembled a professional data collection and annotation team that gathered and processed various tutorials from the internet and also collaborated with artists to customize high-quality painting process data. The datasets vary in size from 50 to 10,000 entries, totaling over 24,000. The first ten tasks have data in 9 frames, while the rest have 4 frames, arranged into 3x3 and 2x2 grids for training purposes. We used GPT-4o to label all datasets and describe each frame.

## 4 Experiment

### 4.1 Experimental Setting

**Setup.** We implemented MakeAnything based on the pre-trained Flux 1.0 dev. We replaced the Adam optimizer with the CAME optimizer, and experiments showed that this setup achieved better generation quality. During the training phases of Asymmetric LoRA and the ReCraft model, the resolution was set to 1024, LoRA rank was 64, learning rate was 1e-4, and batch size was 2. Asymmetric LoRA and the ReCraft model were trained for 40,000 steps and 15,000 steps, respectively.

**Baselines.** In the Text-to-Sequence task, we compare our approach with state-of-the-art baseline methods, namely ProcessPainter Song et al. (2024a), Flux 1.0 AI (2024), and the commercial API Ideogram Ideogram (2023). We categorize the test prompts into two types: painting and others, because some baselines only support painting. In the Image-to-Sequence task, our baselines are Inverse Painting Chen et al. (2024) and PaintsUndo Team (2024), which are capable of predicting the creation process of a painting.

**Evaluation Metrics.** A good procedural sequence needs to be coherent, logical, and useful; however, evaluating procedural sequence generation and its rationality lacks precedents. We employ the CLIP Score to assess the text-image alignment of the generated results. Additionally, we evaluate the coherence and usability of the generated results using GPT-4o and human evaluations. Specifically, we meticulously design the input prompts for GPT-4o and scoring rules to align with human preferences. For comparison, we concatenate outputs from all methods and prompt GPT-4o to identify the best results across different criteria.

### 4.2 Experimental Results

Fig. 4(a) showcases the results of generating process sequences from textual descriptions. Benefiting from high-quality datasets, a robust pre-trained model, and an innovative method design, MakeAnything consistently produces high-quality and logically coherent process sequences. Table 1 presents the quantitative evaluation results of MakeAnything across 21 tasks, including scores from GPT and human assessments, with 20 sequences generated per task.

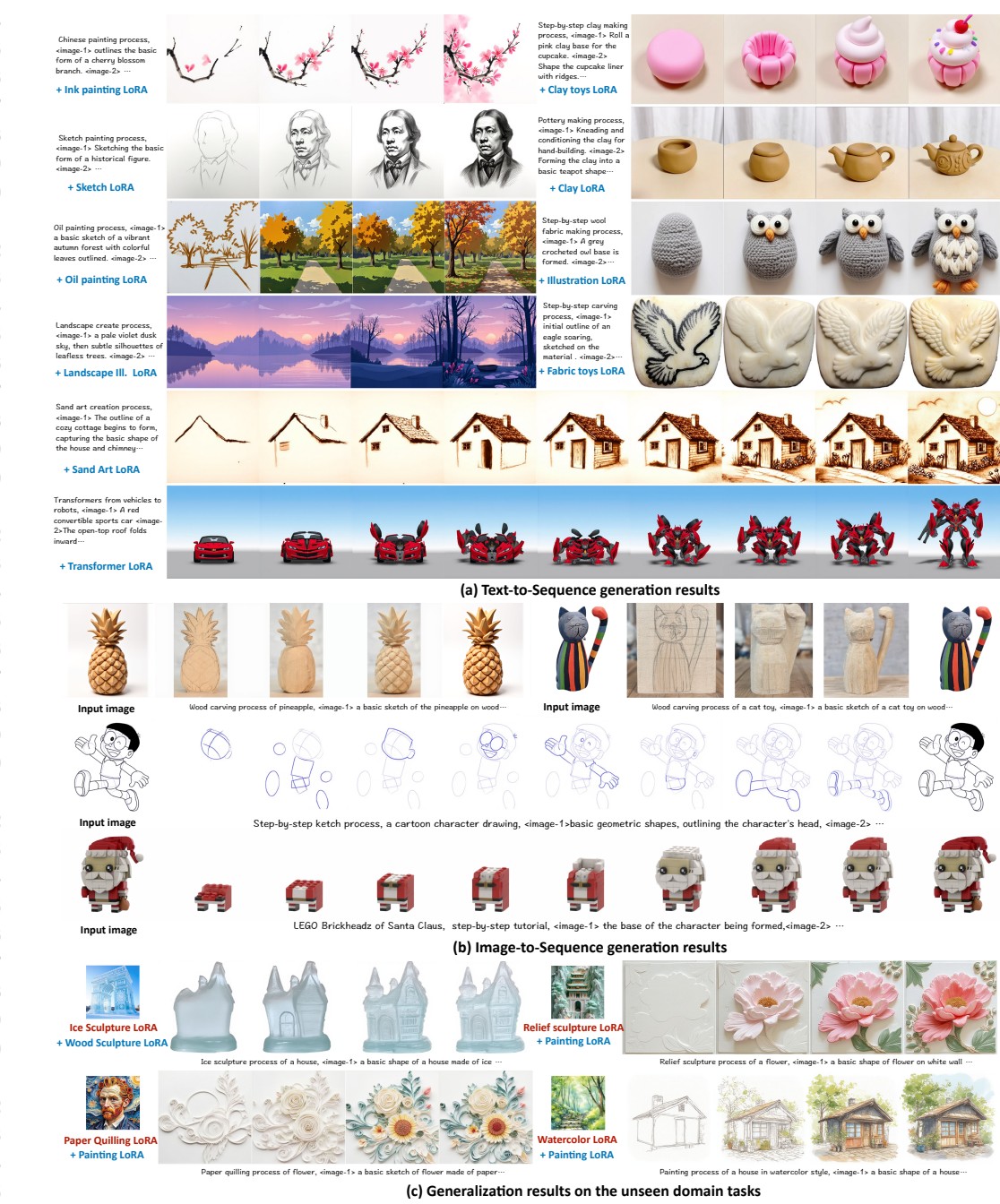

Figure 4: Generation results of MakeAnything. From top: **Text-to-Sequence** outputs conditioned on textual prompts; **Image-to-Sequence** reconstructions via ReCraft Model; **Unseen Domain** generalization combining procedural LoRA (blue) with stylistic LoRA (red).

Fig. 4(b) highlights the model's ability to generate process sequences conditioned on input images. The results indicate a high degree of alignment between the generated sequences and the original image content. This showcases the model's capacity to interpret complex visual inputs and reconstruct logically consistent creation processes, enabling its application in diverse fields such as reverse engineering and educational tutorials.

Fig. 4(c) shows the results of MakeAnything in unseen domains. We collected various LoRAs from the Civitai Civitai (2025) website, including watercolor, relief, ice sculpture, and paper quilling art,

Table 1: Combined evaluation of procedural sequence generation across tasks. G = GPT score, H = Human, C = CLIP.

| Task | Align (G\|H\|C) | Coher (G\|H) | Usab (G\|H) | Task | Align (G\|H\|C) | Coher (G\|H) | Usab (G\|H) |
|---|---|---|---|---|---|---|---|
| Painting | 4.50\|4.27\|34.24 | 4.80\|3.98 | 4.60\|4.13 | Sketch | 4.10\|3.97\|29.35 | 4.70\|4.11 | 4.10\|4.13 |
| Sand Art | 4.20\|4.30\|31.82 | 4.70\|4.12 | 4.30\|4.18 | Portrait | 4.25\|4.28\|33.84 | 5.00\|4.28 | 4.05\|4.33 |
| Icon | 3.45\|4.33\|31.46 | 3.50\|4.17 | 3.15\|4.25 | Landscape Ill. | 4.55\|4.28\|32.25 | 4.85\|3.95 | 4.50\|4.12 |
| Illustration | 3.12\|4.17\|31.68 | 3.40\|4.07 | 2.45\|4.07 | LEGO | 4.60\|4.32\|34.40 | 4.90\|4.15 | 4.75\|4.00 |
| Transformer | 4.75\|4.30\|33.03 | 4.90\|4.23 | 4.75\|4.15 | Cook | 3.20\|4.21\|34.41 | 4.25\|4.03 | 3.65\|3.90 |
| Clay Toys | 4.30\|4.17\|35.25 | 4.50\|4.30 | 4.20\|4.30 | Pencil Sketch | 3.85\|4.33\|34.44 | 4.50\|4.20 | 3.80\|4.25 |
| Chinese Painting | 4.80\|4.37\|33.46 | 4.90\|4.22 | 4.70\|4.33 | Fabric Toys | 4.35\|4.30\|32.83 | 4.60\|4.08 | 4.40\|4.30 |
| Oil Painting | 4.90\|4.30\|37.30 | 4.95\|4.17 | 4.85\|4.20 | Wood Sculpture | 4.65\|4.32\|33.83 | 4.85\|4.23 | 4.65\|4.08 |
| Clay Sculpture | 4.30\|4.17\|35.25 | 4.50\|4.30 | 4.20\|4.30 | Brush Modeling | 4.20\|4.33\|32.27 | 4.15\|4.03 | 4.05\|4.25 |
| Jade Carving | 4.90\|4.28\|32.93 | 4.85\|4.12 | 4.75\|4.00 | Line Draw | 4.10\|4.20\|30.76 | 4.20\|3.97 | 3.90\|4.08 |
| Emoji | 3.75\|4.25\|34.20 | 3.60\|4.17 | 3.80\|4.18 | | | | |

and combined them with our procedural LoRA. It is evident that MakeAnything demonstrates quite impressive generalization capabilities, despite not having been trained on these creative processes.

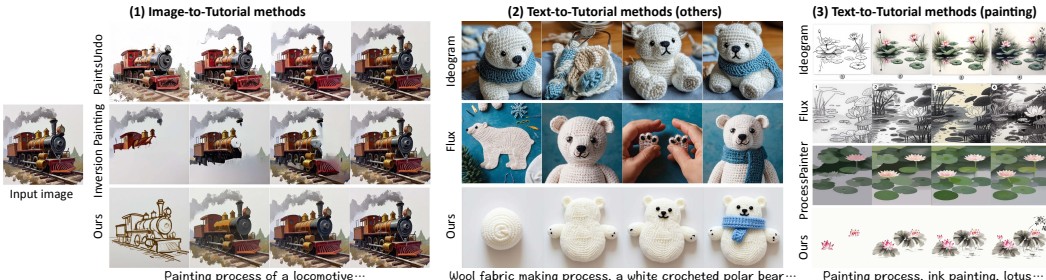

Figure 5: MakeAnything produces more logically consistent sequences compared to baseline methods.

## 4.3 COMPARATION AND EVALUATION

This section consolidates the comparative evaluations of our method against baseline approaches on 50 sequence groups. Fig. 5(a) and (b), demonstrate that MakeAnything produces higher quality procedural sequence with superior logic and coherence, unlike the baseline methods which struggle with consistency. Fig. 5(c) compares the ReCraft model to a baseline, highlighting our method's training on diverse real data, resulting in varied and authentic creative processes. Quantitative results in Fig. 6 confirm MakeAnything's superiority in text-image alignment, coherence, and usability.

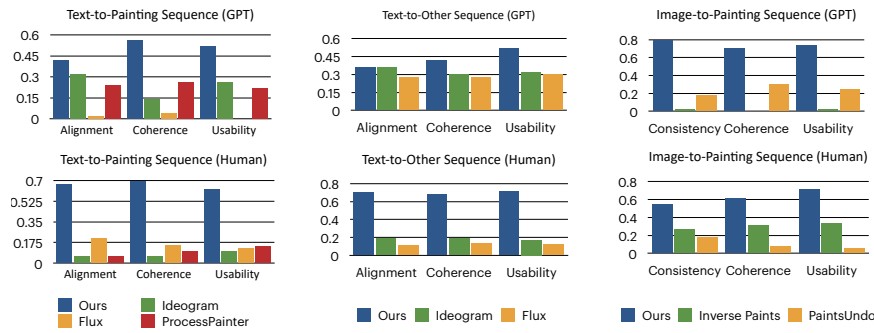

Figure 6: Comparison results on three tasks, evaluated by GPT and humans respectively.

## 4.4 USER STUDY

To comprehensively evaluate MakeAnything's effectiveness, we conducted a user study comparing our method against baselines. Participants rated sequences across four metrics: Alignment (text-image similarity), Coherence (logical step progression), Usability (practical value), and Consistency (consistency between image condition). As shown in Fig. 6, MakeAnything demonstrates comprehensive superiority across all metrics.

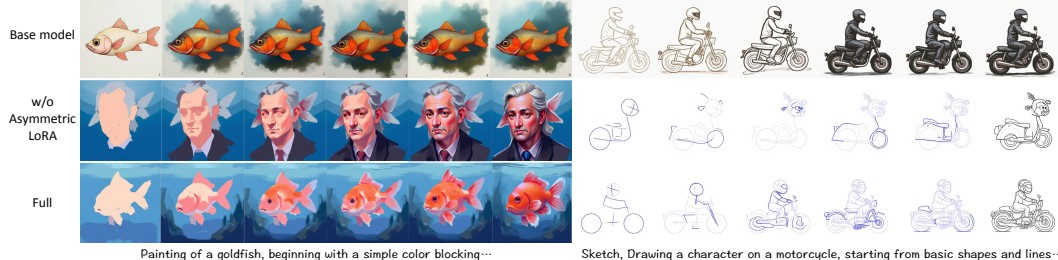

Figure 7: Ablation shows that the full settings yield the best prompt-following ability and sequence coherence, while removing components leads to noticeable performance drops.

## 4.5 ABLATION STUDY

In this section, we conducted ablation experiments on asymmetric LoRA, and Fig. 7 compares the results of portrait and sketch tutorial generation task. The former was trained on 50 portrait painting sequences, while the latter was trained on 300 cartoon character sketch sequences. While the base model produces coherent text but fails in step-by-step synthesis, standard LoRA exhibits severe overfitting on small datasets with imbalanced class distributions—yielding plausible steps but compromised text-image alignment. Our method achieves both procedural rationality and text-image alignment by leveraging knowledge from large-scale pretraining. Quantitative results across more tasks (Table 2) further validate these findings.

Table 2: Ablation Study Results Using GPT Evaluation and CLIP Score.

| Model | Task | Alignment(G \| C) | Coherence | Usability | Task | Alignment(G \| C) | Coherence | Usability |
|---|---|---|---|---|---|---|---|---|
| Base Model | Portrait | 3.75\| 29.78 | 3.45 | 3.35 | Lego | 4.00\| 28.29 | 3.55 | 3.95 |
| | Wood Sculpture | 3.25\| 35.29 | 2.95 | 2.65 | Icon | 3.25\| 32.67 | 3.35 | 3.95 |
| | Fabric toys | 3.55\| 32.95 | 4.00 | 3.85 | Sand Art | 4.05\| 31.75 | 3.85 | 3.80 |
| | Sketch | 3.85\| 29.12 | 3.55 | 3.30 | Oil Painting | 3.65\| 31.05 | 3.70 | 3.70 |
| | Zbrush Modeling | 4.25\| 29.11 | 3.25 | 3.25 | Ink Painting | 3.55\| 31.16 | 3.35 | 3.95 |
| w/o Asymmetric LoRA | Portrait | 4.25\| 31.08 | 4.50 | 4.15 | Lego | 4.20\| 32.69 | 3.85 | 4.45 |
| | Wood Sculpture | 3.55\| 31.05 | 4.35 | 3.75 | Icon | 3.80\| 30.77 | 3.65 | 3.90 |
| | Fabric toys | 3.75\| 30.72 | 3.15 | 3.20 | Sand Art | 3.85\| 30.37 | 3.85 | 4.00 |
| | Sketch | 4.55\| 32.84 | 3.70 | 3.55 | Oil Painting | 4.05\| 32.53 | 3.75 | 3.35 |
| | Zbrush Modeling | 4.15\| 28.00 | 3.80 | 3.40 | Ink Painting | 3.65\| 29.41 | 3.70 | 3.75 |
| Full | Portrait | 4.55\| 32.95 | 4.75 | 4.25 | Lego | 4.35\| 30.01 | 4.15 | 4.65 |
| | Wood Sculpture | 4.25\| 33.89 | 3.80 | 4.05 | Icon | 3.75\| 30.49 | 3.45 | 4.05 |
| | Fabric toys | 4.40\| 32.01 | 4.25 | 4.35 | Sand Art | 4.40\| 30.24 | 4.20 | 4.35 |
| | Sketch | 3.95\| 31.45 | 3.60 | 3.75 | Oil Painting | 3.95\| 29.67 | 3.65 | 3.55 |
| | Zbrush Modeling | 4.55\| 32.48 | 3.95 | 3.55 | Ink Painting | 4.30\| 33.82 | 3.95 | 4.15 |

## 5 LIMITATIONS AND FUTURE WORK

The current grid-based composition strategy in MakeAnything introduces two inherent limitations: constrained output resolution (max 1024×1024) and fixed frame count (up to 9 steps). We plan to address these limitations in future work, enabling arbitrary-length sequence generation with high-fidelity outputs.

## 6 CONCLUSION

We introduced MakeAnything, a novel framework for generating high-quality process sequences using the DiT model with LoRA fine-tuning. By leveraging multi-domain procedural dataset and adopting an asymmetric LoRA design, our approach effectively balances generalization and task-specific performance. Additionally, the image-conditioned plugin enables controllable and interpretable sequence generation. Extensive experiments demonstrated the superiority of our method across diverse tasks, establishing a new benchmark in this field. Our contributions pave the way for further exploration of step-by-step process generation, opening up exciting possibilities in computer vision and related applications.

## CODE OF ETHICS

The authors have read and acknowledge adherence to the ICLR Code of Ethics.

## ETHICS STATEMENT

The datasets used in this work include newly curated materials such as drawing tutorials, process demonstrations, and other publicly shared content. These data sources do not involve sensitive human information and therefore pose no privacy or ethical risks.

## REPRODUCIBILITY STATEMENT

We have fully open-sourced our models, training code, and inference code, enabling complete reproducibility of the results reported in this paper. All hyperparameters, architecture details, and evaluation metrics are documented. With the released dataset, model checkpoints, and scripts, researchers can replicate and extend our findings without ambiguity.

## USE OF LARGE LANGUAGE MODELS

We only used large language models such as GPT-4 and GPT-5 to assist with English grammar refinement and error correction at the writing stage. All technical content—including method design, experimental setup, and quantitative results—was independently conceived, implemented, and verified by the authors. Large language models were not used to modify any experimental data or code. This guarantees the scientific integrity and originality of this work.

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

# A APPENDIX

## A.1 IMPLEMENTATION DETAILS OF THE GPT4-O EVALUATION.

In the GPT-4-o evaluation process, we tailor distinct evaluation metrics for different tasks, ensuring both direct scoring and selective ranking are covered to suit the task's nature.

### A.1.1 DIRECT SCORING EVALUATION (FOR PROCEDURAL SEQUENCE GENERATION AND ABLATION STUDIES)

The assistant evaluates a sequence of images depicting a procedural process with criteria such as:

- **Accuracy:** Measures content alignment with the provided prompt, scored from 1 (not accurate) to 5 (completely accurate).
- **Coherence:** Assesses logical flow from 1 (disjointed) to 5 (seamless progression).
- **Usability:** Rates helpfulness for understanding the procedure from 1 (not helpful) to 5 (highly helpful).

Scores are output in JSON format, for example:

```
{
  "Accuracy": 4,
  "Coherence": 5,
  "Usability": 4
}
```

### A.1.2 SELECTIVE RANKING EVALUATION (FOR USER STUDY COMPARISONS)

This evaluation compares multiple images from different models, ranking them by:

- **Accuracy:** Which image best represents the prompt?
- **Coherence:** Which image shows the clearest, most logical process?
- **Usability:** Which image offers the most helpful visual guidance?

Rankings are provided from 1 (best) to 4 and outputted in JSON format, e.g.,

```
{
  "Accuracy": 1,
  "Coherence": 2,
  "Usability": 3
}
```

**Example of Task Prompt and Evaluation:** Prompt: "This image shows the process of creating a handmade sculpture." Images: [Upload images of models 1, 2, 3, and 4] Evaluation: The assistant ranks the models for Accuracy, Coherence, and Usability in JSON format. This evaluation merges qualitative and quantitative assessments to determine the effectiveness of the images generated by GPT-4-o models.

## A.2 MORE RESULTS

Fig 8-11 show more generation results of MakeAnything. Table 3-6 display the raw data from GPT evaluations and human assessments.

Table 3: Compare with Text-to-Sequence methods (GPT)

| Category | Methods | Alignment | Coherence | Usability |
|----------|---------|-----------|-----------|-----------|
| Painting | Processpainter | 0.24 | 0.26 | 0.22 |
|          | Ideogram | 0.32 | 0.14 | 0.26 |
|          | Flux | 0.02 | 0.04 | 0.00 |
|          | Ours | **0.42** | **0.56** | **0.52** |
| Others   | Ideogram | 0.36 | 0.30 | 0.32 |
|          | Flux | 0.28 | 0.28 | 0.30 |
|          | Ours | **0.36** | **0.42** | **0.38** |

Table 4: Compare with Image-to-Sequence methods (GPT)

| Category | Methods | Consistency | Coherence | Usability |
|----------|---------|-------------|-----------|-----------|
| Painting | Inverse Paints | 0.02 | 0.00 | 0.02 |
|          | PaintsUndo | 0.18 | 0.30 | 0.24 |
|          | Ours | **0.80** | **0.70** | **0.74** |

Table 5: Compare with Text-to-Sequence methods (Human)

| Category | Methods | Alignment | Coherence | Usability |
|----------|---------|-----------|-----------|-----------|
| Painting | Processpainter | 0.06 | 0.10 | 0.14 |
|          | Ideogram | 0.06 | 0.06 | 0.10 |
|          | Flux | 0.21 | 0.15 | 0.13 |
|          | Ours | **0.67** | **0.69** | **0.63** |
| Others   | Ideogram | 0.19 | 0.19 | 0.17 |
|          | Flux | 0.11 | 0.13 | 0.12 |
|          | Ours | **0.70** | **0.68** | **0.71** |

Table 6: Compare with Image-to-Sequence methods (Human)

| Category | Methods | Consistency | Coherence | Usability |
|----------|---------|-------------|-----------|-----------|
| Painting | Inverse Paints | 0.27 | 0.31 | 0.33 |
|          | PaintsUndo | 0.18 | 0.08 | 0.06 |
|          | Ours | **0.55** | **0.61** | **0.61** |

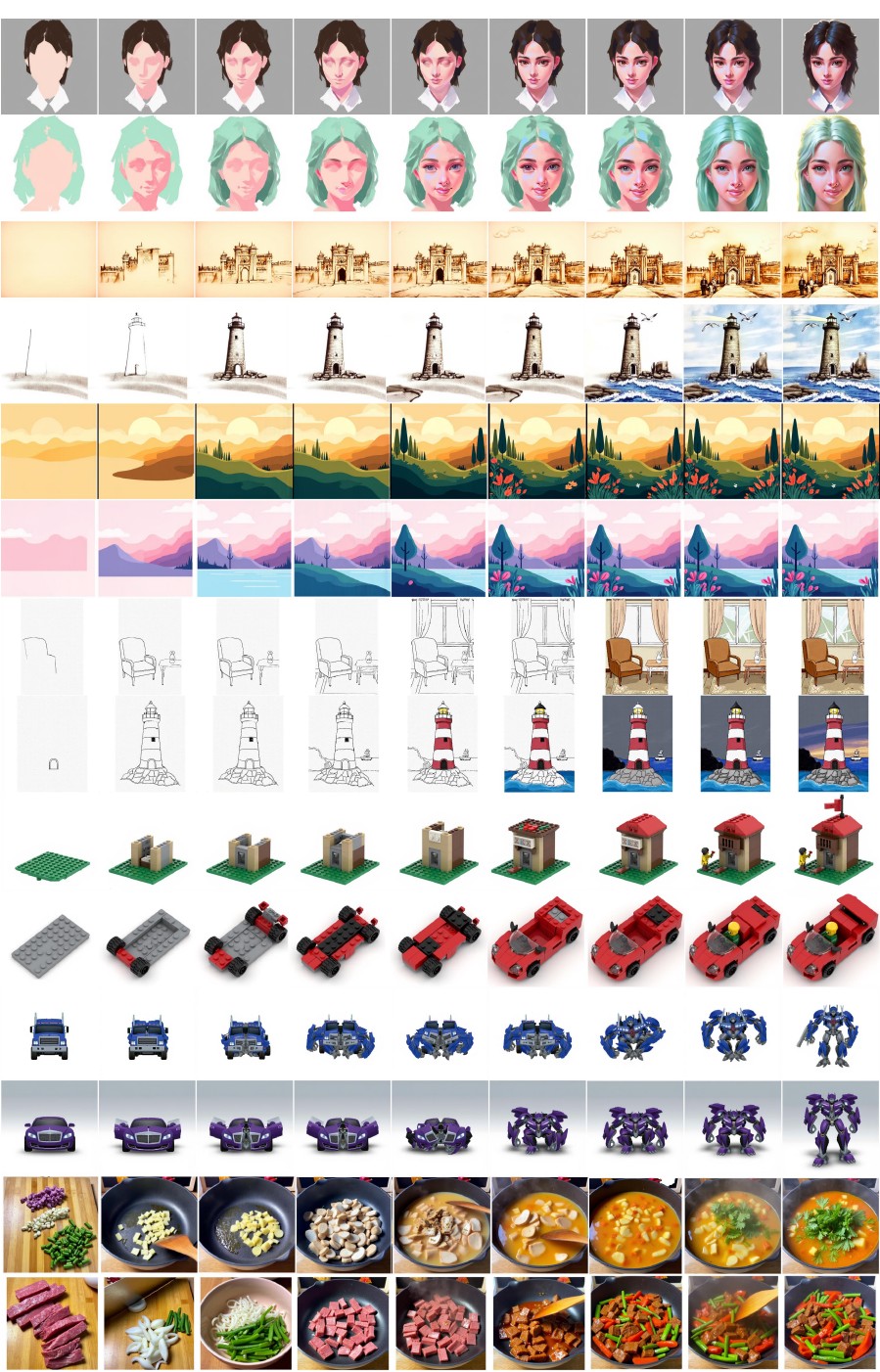

Figure 8: More generation results. From top to bottom, they are portrait, Sand Art, landscape illustration, painting, LEGO, transformer, and cook respectively.

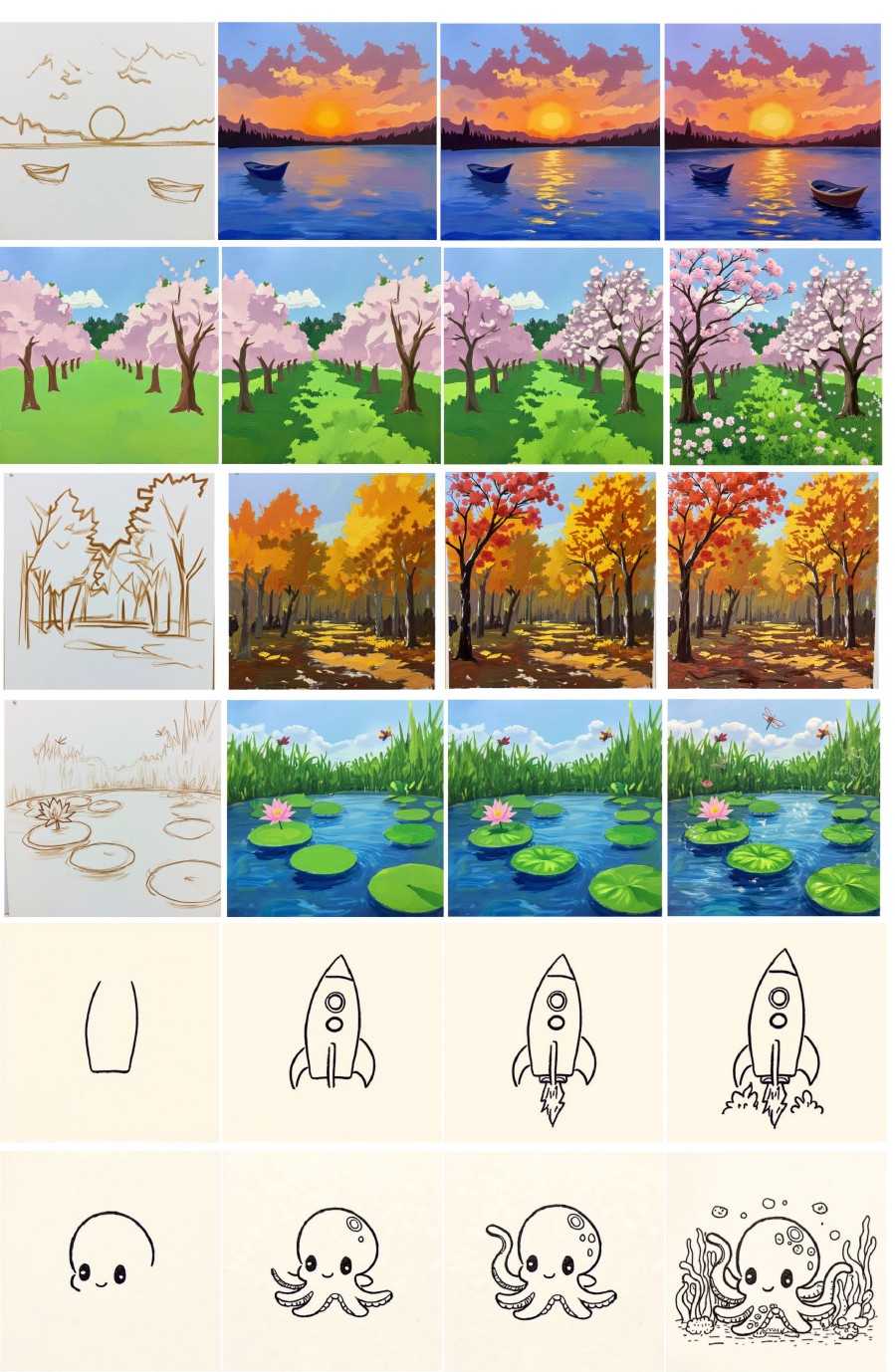

Figure 9: More generation results. From top to bottom, they are oil painting and line draw.

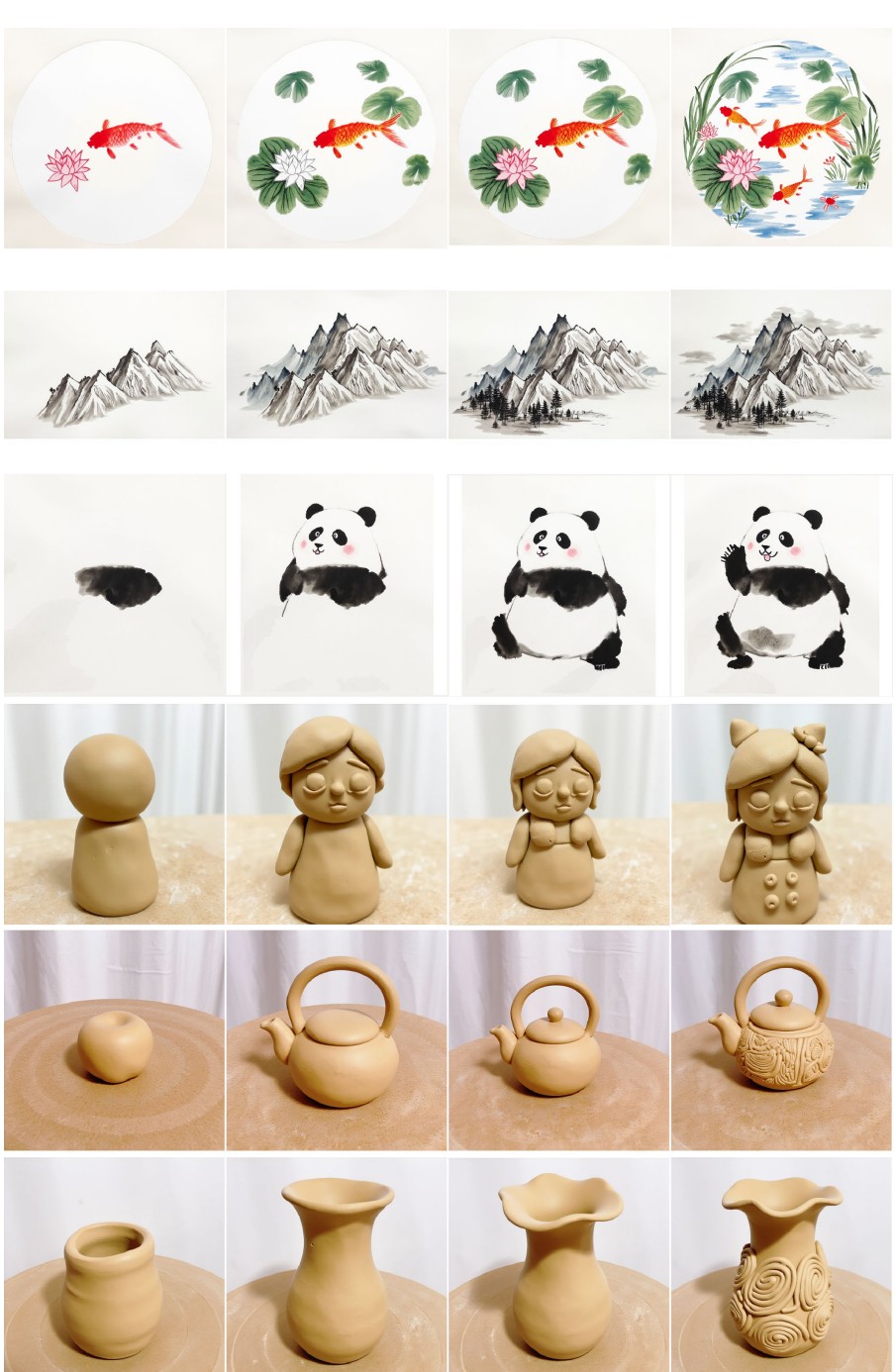

Figure 10: More generation results. From top to bottom, they are ink painting and clay sculpture.

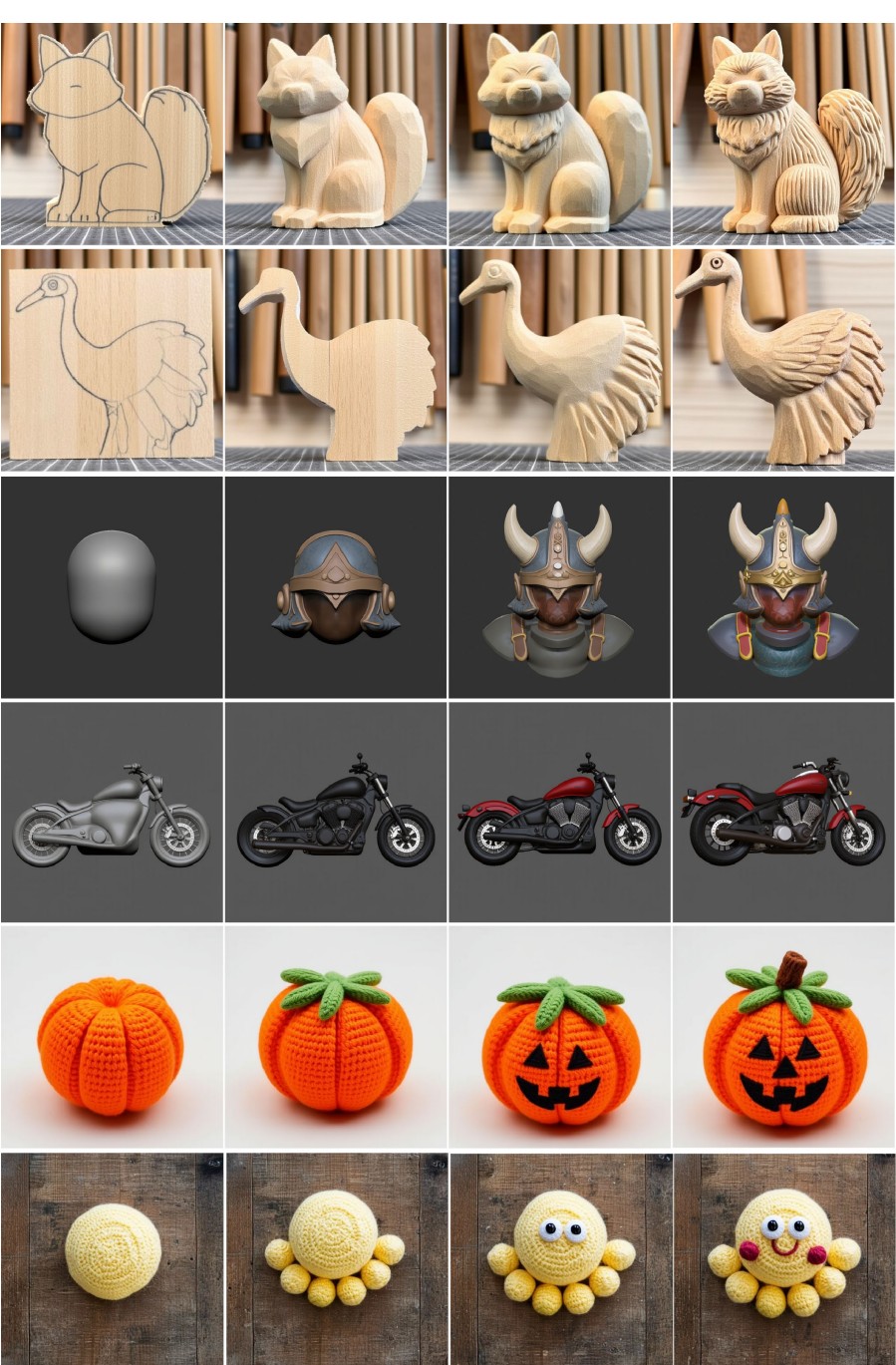

Figure 11: More generation results. From top to bottom, they are wood sculpure, Zbrush, and fabric toys.

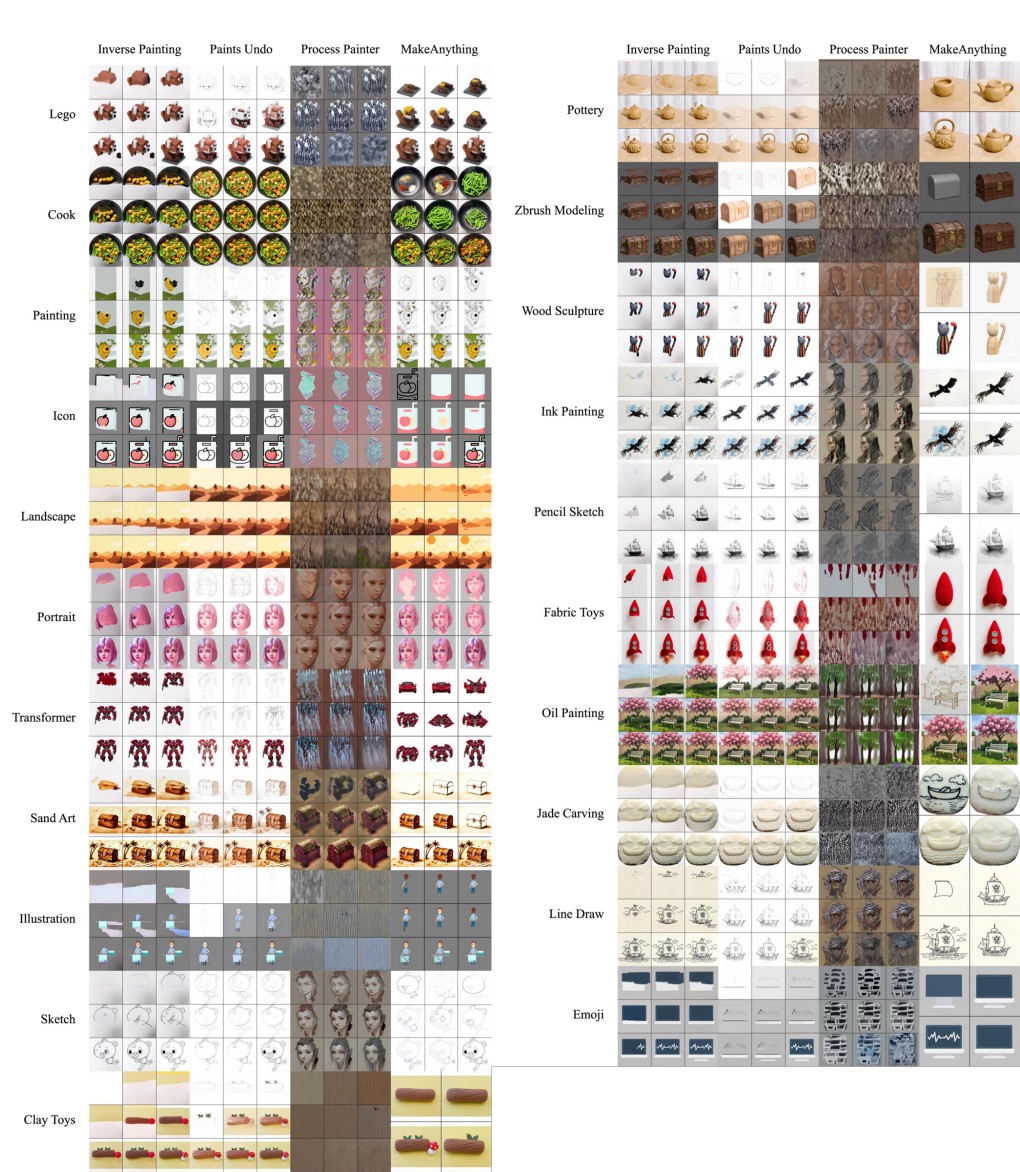

Figure 12: More generation results of MakeAnything and baseline methods.

