# OpenReview forum: "MakeAnything: Harnessing Diffusion Transformers for Multi-Domain Procedural Sequence Generation"
_ICLR.cc/2026/Conference — ICLR 2026 Conference Withdrawn Submission_

### Official Review · Reviewer_Wwhb · 2025-10-21

**Soundness:** 2
**Presentation:** 2
**Contribution:** 3
**Rating:** 4
**Confidence:** 4

**Summary:**

The paper presents a comprehensive framework for generating step-by-step procedural sequences from either text prompts or a single image. It first comes up with a large, multi-domain open-source dataset covering 21 tasks with over 24,000 procedural sequences. Then it fine-tunes a model on these tasks based on Flux using a self-designed asymmetric LoRA method. It also introduces the ReCraft Model for data-efficient, image-conditioned process reconstruction. The fine-tuned model performs good, and is much better than baselines on tasks of text-to-painting, text-to-others and image-to-others under both GPT and human rating.

Although the idea is interesting, the persuasiveness of the evaluation is not as much. The evaluation is messy and lack of details. I suggest to reject the paper if the authors do not offer a clearer version.

**Strengths:**

1: The paper introduces a novel domain of step-by-step procedural sequences generation, and creates a large dataset for the domain.

2: The paper introduces a new method, asymmetric LoRA, to handle the new task.

3: The proposed method is much better than baselines on tasks of text-to-painting, text-to-others and image-to-others under both GPT and human rating.

**Weaknesses:**

1: The paper only lists the implementation details of GPT evaluation, and the vital details of human evaluation, for example, candidate distribution, candidate selection, how to ensure blindness, evaluation quality control etc. are missing.

2: In baseline comparison, the paper only reports a global average for top-1 preference selection of alignment, coherence and usability. GPT may have position bias that favours the first given candidate, while humans may have other forms of bias given the experimental detail being unknown. Also the three sub-term metric seems a little limited, especially for "usability". At least some forms of "rating" and "overall rating" shall be taken into consideration.

Also the evaluation does not distinguish between "seen domains" and "unseen domains". This distinguish shall be vital as it reflects how much proportion the improvement is from overfit/generalization .

3: GPT-4o is used both for data labeling/descriptions and evaluation, this may cause bias as the model may learn to generate images that favours specific modes of GPT-4o description.

4: The paper involves a 2-phase training, while the LoRA model trained on the first phase is merged with the Flux base model to form the base model of the second phase. However, the paper does not describe how they are merged in 3.3 and 3.4.

**Questions:**

1: What's the details of human evaluation?

2: The GPT evaluation and human evaluation diverge by quite a lot. Do you have example showcases that when and why could the preferences be different?

---

### Official Review · Reviewer_fVqP · 2025-10-22

**Soundness:** 3
**Presentation:** 3
**Contribution:** 3
**Rating:** 4
**Confidence:** 3

**Summary:**

The paper introduces MakeAnything, a framework for generating multi-step procedural tutorials from either text or image inputs. To support this goal, the authors curate a multi-domain dataset covering 21 tasks with over 24,000 annotated procedural sequences. Building upon Diffusion Transformers (DiT), MakeAnything learns to synthesize coherent and visually consistent creation processes across diverse domains. In the first stage, the model employs asymmetric LoRA fine-tuning to balance cross-domain generalization and task-specific adaptation for text-to-process generation. In the second stage, the authors introduce ReCraft, an image-conditioned extension that reconstructs creation steps from a single input image by concatenating clean latent tokens from the final frame with noisy tokens through multimodal attention. Their experiments show that MakeAnything outperforms existing baselines (Flux, ProcessPainter, Ideogram) in alignment, coherence, and usability, as validated through both GPT-4o evaluations and human studies. Overall, the paper presents a scalable and generalizable framework for procedural sequence generation.

**Strengths:**

1. The paper addresses the difficult problem of multi-domain procedural generation, which requires maintaining both logical coherence and visual consistency across multiple steps. The introduction of ReCraft further extends the capability from text-to-process to image-conditioned process generation with minimal architectural modifications.
2. The authors curate a new multi-domain dataset of 21 tasks and 24K sequences. It provides a useful resource for studying procedural generation and may help establish a baseline for future work.

**Weaknesses:**

1. Despite careful engineering and a well-organized framework, the paper largely combines and adapts existing ideas rather than introducing a fundamentally new concept. The use of asymmetric LoRA and conditional flow matching loss directly follows prior work. The proposed serpentine sequence layout is more of an engineering trick. Overall, the contribution leans toward system integration and empirical demonstration rather than major advancement.
2. The evaluation lacks several standard quantitative metrics widely used in image generation, such as FID, IS, or KID, which are essential for assessing visual fidelity and diversity. This raises concerns of potential concerns over generation quality.
3. The paper mainly relies on CLIP-based alignment and subjective scores (from GPT-4o and human raters), which do not fully capture image quality or realism. Moreover, the details of these evaluations are insufficiently specified—it remains unclear how many raters participated, whether GPT-4o scoring was calibrated or verified, and how consistency across domains was ensured. As a result, the experimental validity and reproducibility of the reported improvements are questionable.
4. The baseline comparison is relatively weak. The paper mainly contrasts MakeAnything with one prior academic work (ProcessPainter), a commercial system (Ideogram), and Flux. These baselines are either methodologically different or not specifically designed for procedural sequence generation, making the reported improvements difficult to interpret.
5. The task of procedural sequence generation is closely related to image editing and instruction-conditioned visual generation, yet the paper does not clarify how its diffusion-based approach compares to stronger sequence-based or autoregressive models (e.g., GPT-style multimodal transformers).

**Questions:**

1. The paper currently relies on CLIP-based alignment and GPT-4o/human evaluations, but omits standard quantitative metrics such as FID, IS, or KID that are commonly used to assess visual fidelity and diversity. Could the authors clarify why these metrics were excluded, and whether they were tested during development? Additionally, please provide more details about the subjective evaluation setup—number of raters, prompt templates, calibration, and inter-rater consistency—to ensure the reported results are reproducible and statistically meaningful.

2. Given that procedural sequence generation is closely related to sequence modeling and multimodal reasoning, how do the authors justify selecting a diffusion transformer over autoregressive or GPT-style models? Please clarify what concrete advantages (e.g., temporal coherence, controllability, sample diversity) diffusion brings in this context and whether any comparative experiments were conducted.

**Details Of Ethics Concerns:**

None.

---

### Official Review · Reviewer_wrew · 2025-11-01

**Soundness:** 2
**Presentation:** 3
**Contribution:** 3
**Rating:** 4
**Confidence:** 3

**Summary:**

This paper introduces *MakeAnything*, a new framework for generating multi-domain, step-by-step procedural sequences. The authors have curated and proposed a new, large-scale procedural dataset containing over 24K sequences across 21 distinct domains. The work presents three core contributions: a new, large-scale dataset of over 24,000 procedural sequences across 21 distinct domains; the "MakeAnything" text-to-process model, which uses a Diffusion Transformer (DiT) by mapping temporal sequences to a 2D "Serpentine Sequence Layout" to leverage pre-trained spatial reasoning; and the "ReCraft" image-to-process model, which reverse-engineers a static image into a plausible creation sequence using a lightweight conditioning mechanism. The framework also introduces an "Asymmetric LoRA" that balances shared knowledge with task-specific adaptation.

**Strengths:**

The proposed "Serpentine Sequence Layout" leverage the DiT's pre-trained spatial in-context learning capabilities to handle the procedural image sequences generation task, which is a creative and resourceful solution. The Asymmetric LoRA architecture (shared $A$, task-specific $B_i$), while inspired by prior work, is a novel and well-suited application for the difficult problem of multi-domain, data-imbalanced generative tasks. The paper is well-written, with clear problem statements, distinct contributions, and an architectural overview in Figure 2 that makes the model easy to understand. The curated 24k+ sequence dataset is itself a major contribution to the community that will enable future research.

**Weaknesses:**

1. The "Serpentine Layout," while clever, is a fundamental limitation. It hard-codes the sequence length to 4 or 9 steps, which implies the model cannot generate a 6-step or 10-step process. This severely limits practical utility and contradicts the "MakeAnything" title and its goals of general procedural generation.
2. The ablation study in Sec 4.5 is confusing and may be based on a weak baseline. It is unclear what "w/o Asymmetric LORA" represents (is it a per-task LoRA?). What is the "Base model"? If it's the original Flux model, it was never trained on procedural data, so its failure is expected and uninformative.
3. The citation format is strange in the whole article (The brackets have disappeared).

**Questions:**

1. Can the authors comment on the 4/9-step hard constraint from the "Serpentine Layout"? How is a user expected to generate a process of a different length, such as 6 steps, and does this not severely limit the model's practical utility?
2. Can the authors please clarify the baselines in the ablation study (Sec 4.5)? Specifically, what does "w/o Asymmetric LORA" and "Base model" represent, and was a baseline using **a standard LoRA trained on the entire aggregated 24k dataset** ever run? This comparison seems essential to prove the value of the asymmetric design.
3. For the ReCraft model, can the authors provide quantitative (GPT-4o or Human) scores for its performance on non-painting domains (e.g., LEGO, Sketch) to substantiate the multi-domain claims, even if no other baselines are applicable? The author only provide visualization results of the ReCraft model in Figure 4(b) on non-painting domains.

---

### Official Review · Reviewer_8Ccn · 2025-11-03

**Soundness:** 3
**Presentation:** 3
**Contribution:** 2
**Rating:** 4
**Confidence:** 3

**Summary:**

The paper introduces MakeAnything, which is a framework constructed by fine-tuning a DiT (FLUX) model with asymmetric LoRA, enabling it to generate a procedural sequence for different tasks. A multi-domain procedural dataset is proposed to train the model.

**Strengths:**

1. The task that the paper is tackling is interesting and of practical use.

2. The proposed model shows better qualitative results compared to other baselines.

**Weaknesses:**

1. The proposed ReCraft model lacks novelty. According to the descriptions in Section 3.4, it is essentially an image inpainting process where the last cell of the grid (the final image) is fixed while the rest cells are being inpainted - a technique already available in 2022. [1]
In the abstract L42, the authors claim "our ReCraft model enables image-to-process generation through spatiotemporal consistency constraints". However, there is no explicit guarantee of the temporal consistency.

2. The authors introduced CLIP-based text-image alignment, GPT-based, and human-based alignment, coherence, and usability metrics. However, they did not provide any correlation analysis demonstrating that human-based metrics align with GPT or CLIP-based metrics. Simply stashing the results in Tables 1 and 2 makes it hard to understand the correlations.

3. The results for asymmetric LoRA are not consistently better. In Table 2, Sketch, Icon, Oil, and Ink Painting have worse alignment.

4. The proposed Serpentine Sequence Layout lacks an ablation study. It is unclear if it works better than other layouts, such as raster order.

5. There is no detailed information regarding the dataset except for a rough total size. What is the distribution across the proposed 21 tasks?

[1] Lugmayr, Andreas, et al. "Repaint: Inpainting using denoising diffusion probabilistic models." Proceedings of the IEEE/CVF conference on computer vision and pattern recognition. 2022.

**Questions:**

1. For reproducibility and eliminating bias of a specific model (such as GPT-4o), I recommend that the authors perform the evaluation using an open-source MLLM, such as Qwen2.5-VL-72B.

2. Human evaluations lack details. It is unclear, as there seem to be two sets of criteria - direct scoring and ranking.
- How large is the evaluation dataset?
- In A.1.2, does GPT need to output a ranking for each of the samples (as there are 4 samples from different models)?
- How are the numbers in A.2 derived?

3. Regarding the performance dip for painting-related tasks in Table 2 with asymmetric LoRA. Is there a reasonable explanation? Can it be because of the biased distribution across tasks? If so, how about grouping tasks based on their properties and using separate LoRAs for different task groups?

4. In Figure 12, the comparisons in the column on the right are unfair - MakeAnything uses a 2x2 grid while the baselines are all 3x3 grids. Did the author show similar results when doing the user study?

**Details Of Ethics Concerns:**

In L292, the authors describe how the multitask dataset was constructed by gathering and processing various tutorials from the internet. As they did not include details of the data collection process, I have concerns about the copyright and terms of use of the source data; it is unclear whether these were respected during web crawling.

---

### Note · Authors · 2025-11-12

**Comment:**

Thank you very much to all the reviewers for your valuable comments and suggestions. We will carefully consider them and use the feedback to further improve our paper in our next submission.

**Withdrawal Confirmation:**

I have read and agree with the venue's withdrawal policy on behalf of myself and my co-authors.